# P-DROP: POISSON-BASED DROPOUT FOR GRAPH NEURAL NETWORKS

## ABSTRACT

Stochastic processes are widely used in machine learning, yet interacting particle systems—a class of stochastic processes—have seen limited application. In this paper, we leverage an idea from classical interacting particle systems to propose a novel node selection strategy based on Poisson processes. By equipping each node with an independent Poisson clock, our method enables asynchronous and localized updates that preserve structural diversity. This approach introduces not only stochastic but also structure-aware dynamics to graph training.

Recent work has introduced various drop-based techniques such as DropNode, DropEdge, and DropMessage to inject randomness and improve generalization in graph neural networks. Our Poisson-based method offers a principled alternative to these heuristics, yielding competitive or improved performance while grounding the stochasticity in a well-defined process. This work bridges probability theory and graph learning, opening a new avenue for principled stochastic design in GNNs.

## 1 INTRODUCTION

Graphs are widely used in various applications due to their ability to model relationships between entities (nodes) through connectivity. This structural information allows for more expressive representations of complex datasets.

With the rise of machine learning, Graph Neural Networks (GNNs) have gained significant popularity. Along with this rise, several challenges have emerged, particularly the need for effective regularization. Most GNN architectures rely on message passing to aggregate information from a node's local neighborhood in order to capture structural dependencies. While this mechanism is powerful, it can also lead to redundancy and reduced robustness if local interactions are not properly regularized. In contrast to overfitting, which occurs when a model memorizes training data and fails to generalize, here the challenge lies in balancing stochasticity and structure so that local neighborhoods remain informative without overwhelming the model with repeated or highly correlated signals. This motivates the design of dropout-style approaches that regularize local interactions in a principled way.

On the other hand, in probability theory, Interacting Particle Systems are used for many fields, including statistical physics, biology, economics, social sciences, and more. It has some common things with Graph Neural Networks. It utilizes graph structures, which contain vertices and edges, to take advantage from connectivity or relativity between nodes. In this paper, we leverage the method from how Interactive Particle Systems propagate.

For better regularization, from the dropout(Srivastava et al. (2014)), various sampling and propagation control strategies have been proposed(Do et al. (2021), Rong et al. (2020),Fang et al. (2023). In this paper, we introduce a node selection mechanism based on a Poisson process, which probabilistically controls the propagation path in GNNs by leveraging local connectivity patterns.

## 2 BACKGROUND

We introduce a stochastic update mechanism for graph neural networks based on contact process from interacting particle systems. This framework provides a principled way to regulate update sparsity and timing, which supports localized information propagation.

A Poisson process is a fundamental stochastic model for random events occurring over time with a constant average rate $\lambda > 0$. Formally, let $\{N(t) : t \geq 0\}$ be a Poisson process with rate $\lambda$. The number of events in any interval of length $t$ follows a Poisson distribution:

$$\mathbb{P}(N(t) = k) = \frac{(\lambda t)^k}{k!} e^{-\lambda t}, \quad k = 0, 1, 2, \ldots$$

A key property is that the inter-arrival times $T$ between consecutive events are independent and identically distributed exponential random variables with rate $\lambda$:

$$T \sim \text{Exp}(\lambda)$$

and probability density function of $T$ is given by

$$\lambda e^{-\lambda t}, \quad t \geq 0.$$

which has the memoryless property, formally expressed as

$$P(T > s + t | T > s) = P(T > t)$$

indicating that the waiting time does not depend on how much time has already passed.

More detailed explanations of interacting particle systems will be provided in the Appendix.

**Local Clocks and Asynchronous Updates.** In our method, each node $v \in V$ is equipped with an independent Poisson clock with rate $\lambda_v$, which determines when it becomes eligible for update. At each time $t$, we define the active node set as:

$$V_{\text{active}}(t) = \{v \in V : T_v \leq t\}.$$

This asynchronous update scheme ensures that:

- Nodes update sparsely and independently, mitigating simultaneous over-updating.
- Node update frequency can be customized via $\lambda_v$, reflecting structural importance.

The exponential distribution's memoryless property further suits recurrent and decentralized update scheduling.

**Superposition and Indirect Influence.** The superposition property of Poisson processes states that the union of independent Poisson processes is itself a Poisson process. As a result, although a node may not be directly active at time $t$, it may still participate indirectly in the update through neighbors whose clocks have triggered. The aggregated activity ensures that nodes embedded in highly connected or frequently updated regions continue to receive information, even if they are not directly selected.

**Comparison to Uniform Dropout Methods.** Traditional methods such as DropNode or DropEdge randomly suppress nodes or edges uniformly(Fang et al. (2023)), regardless of graph structure. This may inadvertently remove crucial hubs or disconnect important subgraphs, degrading message propagation. In contrast, our Poisson-based strategy preserves structural integrity, as key nodes either update directly or remain involved through their neighbors.

**Relation to Stochastic Models.** Our approach is inspired by stochastic interacting particle systems such as the contact process (Lanchier (2024)), where local Poisson clocks drive the spread of information across space and time. Similarly, in our setting, each node initiates local propagation based on its own stochastic clock, enabling spatially localized and temporally diverse updates.

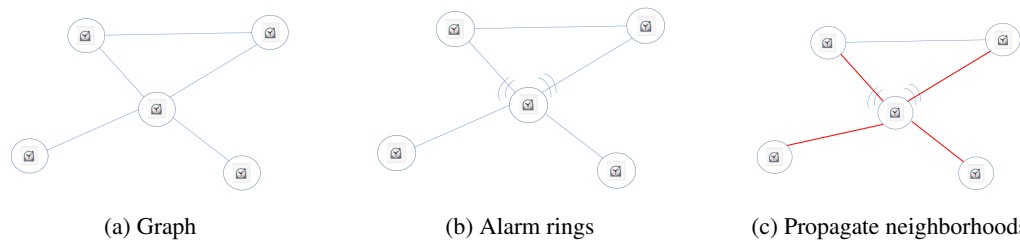

(a) Graph        (b) Alarm rings        (c) Propagate neighborhoods

Figure 1: How it works.

## 2.1 SAMPLING-BASED APPROACHES FOR REGULARIZATION

GNNs are prone to overfitting due to their high model capacity and the sparsity of labeled data. To mitigate this issue, sampling-based regularization methods have been proposed that introduce stochasticity during training.

DropEdge (Rong et al. (2020)) randomly removes a subset of edges in the input graph, weakening structural connectivity and limiting the influence radius of each node. This not only regularizes training but also prevents over-reliance on specific graph connections. More recently, DropMessage (Fang et al. (2023)) introduces randomness at the message-passing level, skipping messages along existing edges with a certain probability. This fine-grained strategy reduces redundancy in aggregated features while preserving the global topology.

While effective, most existing methods rely on uniform random sampling, which may overlook the structural heterogeneity of graphs. In this work, we introduce a Poisson process-based node selection mechanism that leverages local connectivity patterns to probabilistically control propagation, providing a structure-aware form of regularization.

## 3 PROPOSED METHODS

Our proposed idea can be considered as a "degree-aware" method. Also, changing lambda, we can make it more or less sensitive to degree for updates. It will be dealt with in Supplementary.

To be explicit, it is using of Poisson processes for node selection. How it is used is thatto reinterpret dropout in Graph Neural Networks through a Poisson-based mechanism. Traditional dropout methods, such as DropNode or DropEdge, rely on fixed probabilities or heuristics to randomly remove nodes or edges during training. In contrast, our approach introduces a stochastic but structurally-aware alternative by sampling nodes according to a Poisson process, potentially making dropout more adaptive to the underlying graph structure.

Also We explore both of these directions in the subsequent subsections.

## 3.1 POISSON-BASED ALTERNATIVES TO DROPOUT REGULARIZATION IN GNNS

As previously discussed, one common approach to mitigating overfitting and over-smoothing in GNNs is to apply dropout-based regularization techniques.

---

**Algorithm 1** PoissonDropNode with Time-Stepping Simulation.

---

1: **procedure** RUN($T, \Delta t, \lambda$)
2:     $t \leftarrow 0$
3:     **while** $t < T$ **do**
4:         For each node $i$, sample $\tau \sim \text{Exponential}(\lambda)$
5:         **if** $\tau \leq \Delta t$ **then**
6:             Mark node $i$ as active
7:         **end if**
8:         $t \leftarrow t + \Delta t$
9:     **end while**
10: **end procedure**

---

In the proposed algorithm, each node is assigned an independent exponential random variable with rate $\lambda$. Nodes whose sampled values greater than the threshold are selected to form the subgraph used for propagation.

This Poisson-based node selection scheme can be interpreted as a structural alternative to standard Dropout. Unlike uniform random sampling, it allows the selection process to be sensitive to the structural properties of the graph, thereby introducing a form of structure-aware regularization.

### 3.2 DEGREE SENSITIVITY

In the Poisson process, exponential random variables are used to model the inter-arrival times between consecutive events. The rate parameter $\lambda$ determines the expected frequency of events per unit time, with

$$\mathbb{E}[X] = \frac{1}{\lambda}.$$

Assigning a Poisson clock to each node allows us to adjust $\lambda$ according to node degree, thereby controlling how sensitive the update process is to connectivity.

**Proposition 3.1.** *If we use a degree-dependent rate*

$$\bar{\lambda}_i = \lambda\,(1 + \alpha d_i),$$

*for each node $i$ with degree $d_i$, then the expected inter-arrival time of node $i$ is*

$$\mathbb{E}[T_i] = \frac{1}{\bar{\lambda}_i}.$$

*Since this expectation decreases as $d_i$ increases, nodes with higher degree are activated more frequently, making the process degree-sensitive.*

**Proposition 3.2.** *If instead we use*

$$\bar{\lambda}_i = \frac{\lambda}{1 + \alpha d_i},$$

*for each node $i$ with degree $d_i$, then the expected inter-arrival time of node $i$ is*

$$\mathbb{E}[T_i] = \frac{1}{\bar{\lambda}_i} = \frac{1 + \alpha d_i}{\lambda}.$$

*Here, the expectation increases with $d_i$, so nodes with higher degree are activated less frequently, making the process less sensitive to hubs.*

These two formulations allow P-DROP to either accentuate or mitigate the influence of node degrees, depending on the task requirements.

## 4 EXPERIMENTS

We adapted the official implementation of DropMessage from (Fang et al. (2023)), making necessary modifications to suit our method. As a result, our experimental setup remains largely consistent with the original paper, and our results are comparable in terms of trends and performance.

## 4.1 DROP-BASED METHODS ON HOMOPHILIC GRAPHS

We use three widely adopted citation network datasets:

- **Cora**: 2,708 nodes, 5,429 edges, 7 classes.
- **CiteSeer**: 3,327 nodes, 4,732 edges, 6 classes.
- **PubMed**: 19,717 nodes, 44,338 edges, 3 classes.

Each node represents a scientific publication, and edges denote citation links. Node features are bag-of-words representations, and the task is to predict the research category of each node.

We used PyTorch Geometric to implement various dropout-based regularization methods.

| Model | Cora | CiteSeer | PubMed |
|---|---|---|---|
| GCN-Dropout | 83.00 | 72.80 | 80.10 |
| GCN-DropEdge | 80.40 | 71.00 | 80.10 |
| GCN-DropNode | **83.50** | **73.10** | 80.20 |
| GCN-DropMessage | 82.80 | 72.80 | 80.10 |
| GCN-PDROP | 82.70 | 72.60 | **80.30** |
| GAT-Dropout | **83.70** | 71.70 | **79.20** |
| GAT-DropEdge | 81.40 | **72.40** | 78.20 |
| GAT-DropNode | 80.50 | 68.00 | 77.60 |
| GAT-DropMessage | 82.90 | 71.90 | 78.60 |
| GAT-PDROP | 82.50 | 71.80 | **79.20** |
| APPNP-Dropout | 82.45 | 72.20 | 78.64 |
| APPNP-DropEdge | 83.60 | 73.00 | 80.30 |
| APPNP-DropNode | 83.50 | 73.10 | 80.40 |
| APPNP-DropMessage | 82.80 | 72.80 | 80.30 |
| APPNP-PDROP | **83.60** | **73.10** | **80.90** |

Table 1: Node classification accuracy (%) on Cora, CiteSeer, and PubMed. Best results per column are in bold.

Note that we fine-tuned Pdrop, and the reported results are the averaged best scores across 10 fixed seeds.

Table 1 summarizes the final validation and test accuracy for each method across the three benchmark datasets. Overall, the performance differences among the models are relatively small. However, our proposed SGNN method achieved highly competitive results. In particular, SGNN obtained the highest test accuracy on Pubmed and matched or exceeded the performance of DropEdge and DropNode on Cora. While Citeseer results were slightly lower, the model still remained within a close margin. These results indicate that the Poisson-based update mechanism is effective and robust across different graph structures.

## 4.2 EXPERIMENTS ON PROTEIN GRAPHS

We evaluate our framework on the TU PROTEINS dataset, where each protein structure is represented as a graph of amino acid residues with edges encoding spatial proximity. Our architecture consists of two graph convolutional layers (GCN, GAT, or APPNP backbones), followed by global mean pooling and global max pooling. The pooled features are concatenated and passed through a two-layer MLP classifier. We train using the Adam optimizer with a learning rate of 0.001, hidden dimension of 64, dropout rate of 0.3, and weight decay of $10^{-4}$. The dataset is split into 70% training, 15% validation, and 15% testing, and all experiments are run with fixed random seeds for reproducibility.

On PROTEINS (1,113 graphs, average $\approx$ 39 nodes per graph), our method achieves 80–82% accuracy, improving over standard GCN and GAT baselines by approximately 1–3 percentage points. The improvement is consistent across runs and demonstrates that the proposed regularization strategy is beneficial when labels correlate with short-range neighborhoods.

**Discussion.** The initial motivation for P-DROP comes from protein structures: residues with high degree, being more extensively connected, often play a central role in folding stability and functional organization. Protein structures also naturally exhibit local homophily, where residues close in space tend to share functional roles. By perturbing node features stochastically during training, our method reinforces robustness to local variations while preserving essential neighborhood information, which explains the observed performance gains. This perspective further motivated our degree-aware extension of P-DROP, where nodes with larger degrees are activated more frequently, reflecting their structural importance in the molecular graph. Taken together, these results highlight that our approach is particularly effective in domains with locally coherent structures (e.g., proteins).

### 4.3 LIMITATION ON HETEROPHILIC GRAPHS

While P-DROP consistently regularizes local interactions, we observe weaker gains on heterophilic benchmarks. We hypothesize that the mechanism that benefits homophilous graphs—prioritizing frequently activated local neighborhoods—can under-exploit informative long-range signals that are characteristic of heterophily. In tasks where label-relevant cues reside beyond 1–2 hops or require relation polarity/direction, a purely local activation schedule may attenuate effective paths. This does not imply that the graph is uninformative; rather, it highlights a mismatch between a local-interaction prior and a non-local signal structure. We view these results as scope conditions for P-DROP and a guide for extensions that incorporate non-local propagation or edge semantics.

Two practical remedies are (i) augmenting P-DROP with non-local propagation (e.g., personalized-teleport diffusion or multi-hop mixing) so that activation events can access distant evidence, and (ii) learning degree- and edge-aware rates $\lambda_i, \lambda_{ij}$ that adapt activation to hubs, directions, or signed relations. We leave systematic exploration of these variants to future work.

## 5 CONCLUSION

In this project, we introduced a Poisson-based node selection mechanism for Graph Neural Networks to address the over-smoothing problem. By assigning each node an independent Poisson clock, our method enables asynchronous and structure-aware updates, either as a dropout regularization technique or as a subgraph-based training framework. Experiments on standard citation networks show that, despite slower initial performance, our method converges to comparable or superior accuracy in later stages, especially on larger graphs like Pubmed. These results suggest that leveraging stochastic update timing informed by graph structure can offer a promising direction for improving training efficiency and performance in GNNs.

## 6 FUTURE WORK

### 6.1 OVER-SMOOTHING

The initial motivation for this model was to address the over-smoothing problem inherent in deep Graph Neural Networks. While the theoretical framework has been established, the current implementation is still under development due to limited time and computational resources. To advance this direction, we plan to analyze and build upon existing implementations from prior studies, which have explored various strategies to mitigate over-smoothing.

In particular, we observed promising results from our drop-based models, suggesting that the Poisson-based update mechanism may also serve as an effective regularization tool against over-smoothing. These preliminary findings motivate further exploration and refinement of the method, especially in deeper architectures where the issue becomes more pronounced.

# 7 APPENDIX

## A COMPUTATION ANALYSIS

Dropout variants in Graph Neural Networks(e.g., DropEdge, DropNode, DropMessage) often introduce computational overhead, like adjacent matrix reconstruction. In contrast, P-DROP leverages a Poisson-clock mechanism to select active nodes/edges asynchronously, which allows us to reduce redundant computations while maintaining stochasticity. To be more explicit, we assign the clock to each vertex and as time goes by, we use an exponential random variable to get memoryless property. In a glance, it might consume a lot of resources, but in fact, it is opposite.

**Theorem A.1** (Computation Cost of P-DROP). *Let $T > 0$ be the training horizon, and let $\Delta t > 0$ be the discretization step. Suppose that each node $v \in V$ is equipped with an exponential random variable to simulate its Poisson clock. Then the total number of computations required by P-DROP over one epoch is bounded by*

$$\mathcal{C}(T, \Delta t) \ \leq \ 3|V| \cdot \left\lceil \frac{T}{\Delta t} \right\rceil.$$

*Here, the factor $2|V|$ accounts for sampling the exponential random variables and updating the corresponding clocks for all nodes at each step.*

*Proof.* We divide the training horizon $[0, T]$ into $\lceil T/\Delta t \rceil$ time steps of length $\Delta t$. At each step, every node $v \in V$ requires two operations: (1) sampling an exponential random variable to simulate the Poisson clock, (2) comparing time and poisson clocks for each node, and (3) updating the clock state accordingly. Therefore, each step requires at most $3|V|$ operations. Summing over all steps gives

$$\mathcal{C}(T, \Delta t) \leq 3|V| \cdot \left\lceil \frac{T}{\Delta t} \right\rceil.$$

$\square$

One distinctive feature of P-DROP, compared to other dropout schemes, is that the horizon $T$ and step size $\Delta t$ can be explicitly chosen, which enables a balance between accuracy and computational cost. This provides an additional degree of fine-tuning in the training process. By contrast, conventional methods typically rely on repeated reconstruction of adjacency matrices, which is computationally heavy.

## USE OF LARGE LANGUAGE MODELS

In accordance with the ICLR 2026 policy on the use of large language models (LLMs), we disclose that some LLMs were used as a writing and coding assistant during the preparation of this paper. Specifically, the LLM was employed to help polish the presentation of certain sections for clarity and to provide assistance with coding tasks, such as pseudocode drafting and debugging of the `PoissonDropNode` implementation. All outputs from the LLM were carefully reviewed, verified, and, when necessary, edited by the authors to ensure correctness and faithfulness to our intended contributions. All conceptual contributions, theoretical developments, experimental design, and analysis were conducted by the authors. The LLM served only as a supporting tool and is not considered a contributing author.

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
