# OpenReview forum: "P-DROP: Poisson-Based Dropout for Graph Neural Networks"
_ICLR.cc/2026/Conference — ICLR 2026 Conference Withdrawn Submission_

### Official Review · Reviewer_Mxim · 2025-10-17

**Soundness:** 1
**Presentation:** 1
**Contribution:** 1
**Rating:** 2
**Confidence:** 4

**Summary:**

The authors propose to activate nodes in a graph neural network using independent Poisson processes per node. It is established in the literature that allowing for asynchronous message passing can aid in mitigating effects to over-squashing or over-smoothing. The novel contribution is to use a Poisson based distribution instead of other heuristics.

The method is tested on common datasets such as Cora, CiteSeer, PubMed, and the TU Proteins dataset. No clear advantage over SOTA is demonstrated.

**Strengths:**

- The general idea to model node activations using some underlying stochastic process can make sense, and is interesting.
- The degree-aware parametrization is interesting, and investigating this further may make sense.

**Weaknesses:**

- Novelty is limited: The Poisson clock is equivalent to per-node Bernoulli dropout with node-specific rates. Hence, P-DROP is close to non-uniform DropNode.
- In all reported performance benchmarks, error estimates are missing and no advantage can be deduced (although apparently 10 runs were performed).
- Two degree aware variants are discussed to either enhance or mitigate the influence of the graph structure on the node activation, but it does not become clear which is used, and what respective advantages/disadvantages would be. Any ablation on this is missing.
- The algorithm may be computationally expensive, since for every node one needs to sample from the stochastic process. In addition, the resulting masks lead to a re-normalization of information propagation with slower information spread.
- It is claimed that the approach aids in over-smoothing and over-squashing, but there is no quantitative or qualitative analysis showing this.
- Connection to contact-processes is hinted at, but not discussed later in the manuscript.
- Inconsistent writing: For example the acronym SGNN is not introduced and used only in two adjacent sentences.
- Reproducibility is limited; there is no discussion of hyperparameter tuning or code availability.

**Questions:**

The questions follow from the weaknesses:
- How do degree-aware variants improve the results?
- How similar is P-DROP to other variants of DROP.
- Does P-DROP help with over-smoothing and over-squashing? How?
- How big is the computational overhead of P-DROP?

---

### Official Review · Reviewer_1tu7 · 2025-10-31

**Soundness:** 3
**Presentation:** 3
**Contribution:** 3
**Rating:** 6
**Confidence:** 3

**Summary:**

This paper proposes a Poisson distribution-based node selection method for GNN training. This approach probabilistically controls the propagation paths in GNNs based on local connectivity patterns. The proposed method addresses the over-smoothing problem in GNN training. Experiments confirm that in the late stages, the method achieves higher accuracy and improves training efficiency and performance.

**Strengths:**

1. It fills the gap of poor effectiveness and performance of dropout methods in GNN training.
2. The proposed node selection method fully considers the structural features of the graph, rather than relying solely on uniform sampling as in previous work.
3. Experimental results validate the effectiveness of the proposed method.

**Weaknesses:**

1. The paper lacks an in-depth analysis of the proposed method and only presents the conclusion and results.
2. Some of the experimental results in Table 1 are not optimal, and the improvements are extremely limited.
3. The experimental section lacks certain ablation experiments, such as the impact of each node’s λ value on performance and how to determine the λ value for each node.
4. The parameters of the baseline systems in the experiments are not mentioned, such as the dropout parameter in Table 1.

**Questions:**

1. How is the λ value for each node determined?

---

### Official Review · Reviewer_yNo7 · 2025-11-01

**Soundness:** 2
**Presentation:** 1
**Contribution:** 2
**Rating:** 0
**Confidence:** 4

**Summary:**

This paper introduces a new way to incorporate stochasticity in graph neural networks, based on Poisson processes. The nodes are updated asynchronously with messages controlled by a node-wide clock. Experiments are conducted to show the proposed method does not perform worse than existing approaches.

**Strengths:**

The proposed idea of asynchronously updating nodes is interesting.

**Weaknesses:**

- The paper is haphazardly presented with countless grammatical errors,
- There is limited technical innovation in the proposed method.
- The mathematical reasoning is very informal.
- The performance of the proposed method is not better.
- Only outdated, small benchmark datasets are used.

**Questions:**

See weaknesses.

---

### Official Review · Reviewer_auD7 · 2025-11-01

**Soundness:** 2
**Presentation:** 1
**Contribution:** 2
**Rating:** 2
**Confidence:** 3

**Summary:**

The paper introduces P-DROP, a structure-aware alternative to uniform drop-based regularization for GNNs. Instead of indiscriminately removing nodes or edges (as in DropNode or DropEdge), it employs an asynchronous, structure-sensitive node selection scheme driven by independent Poisson clocks. A node becomes eligible for update when its clock “rings,” introducing principled stochasticity grounded in probability theory rather than heuristic randomness.

**Strengths:**

- Proposes a novel, Poisson-clock–based node-selection mechanism for GNN dropout.
- Presents a more principled alternative to uniform, heuristic drop schemes by grounding stochasticity in probability theory.
- The asynchronous, sparse activation design is a promising direction for alleviating over-smoothing in deeper GNNs

**Weaknesses:**

The core idea is clear, but the writing and experimental evaluation need substantial improvement:
- Underspecified math: Key parameters (λ, α) are neither clearly defined nor justified.
- Figure 1 is there but never got mentioned.
- Naming inconsistency: The method is referred to as both “P-DROP” and “SGNN,” causing confusion; standardize terminology.
- Core claim untested: Over-smoothing mitigation is asserted but not evaluated on deep GNNs; include 8–32-layer studies.
- Narrow benchmarks & weak reporting: Results are limited to a single table on Cora/CiteSeer/PubMed. Protein-graph results are brief, not tabulated, and use only GCN/GAT (no GIN).
- No ablations: Lacks tests isolating degree sensitivity (α) from asynchronous timing; add controlled ablations and sensitivity plots.

**Questions:**

Please see the weaknesses. I am also confused about the method. You mention two degree-dependent rate options (Propositions 3.1 and 3.2), which one did you use in the experiments? More details, explanations, and investigation are neede, for example, which case is better under which conditions?

---

### Note · Authors · 2025-11-29

I have read and agree with the venue's withdrawal policy on behalf of myself and my co-authors.